# Second-Generation Neuroendocrine Immunohistochemical Markers: Reflections from Clinical Implementation

**DOI:** 10.3390/biology10090874

**Published:** 2021-09-05

**Authors:** Carl Christofer Juhlin

**Affiliations:** 1Department of Oncology-Pathology, Karolinska Institutet, 17164 Solna, Sweden; christofer.juhlin@ki.se; 2Department of Pathology and Cytology, Karolinska University Hospital, 17176 Stockholm, Sweden

**Keywords:** neuroendocrine tumor, neuroendocrine carcinoma, pathology, immunohistochemistry, marker, second-generation, ISL1, INSM1

## Abstract

**Simple Summary:**

Neuroendocrine tumors are a collection of neoplastic lesions arising in cells with traits similar to hormone-producing and nerve cells with the ability to secrete peptide hormones using an intricate vesicle transportation system. From a clinical standpoint, neuroendocrine tumors are unique in terms of therapeutic modalities, and a correct diagnosis is therefore imperative in order for the patient to obtain the most efficient treatment. In this process, the pathologist can analyze if the tumor cells express Chromogranin A and Synaptophysin, two proteins associated with the regulation of secretory vesicles. Unfortunately, these markers are not always present in neuroendocrine tumors, and non-neuroendocrine tumors may also occasionally express Chromogranin A or Synaptophysin—making the diagnosis difficult to make for certain cases. Recently, three proteins termed ISL1, INSM1 and Secretagogin were found to be selectively expressed in neuroendocrine cells, and subsequent studies have identified their potential as markers of neuroendocrine differentiation in the clinical setting. In this commentary, the benefits of these novel “second-generation” markers are briefly discussed from a clinical context.

**Abstract:**

When analyzing tumors by histopathology, endocrine pathologists have traditionally been restricted to a few key immunohistochemical markers related to secretory vesicles in order to pinpoint neuroendocrine differentiation—most notably Chromogranin A (CGA) and Synaptophysin (SYP). Although proven of great clinical utility, these markers sometimes exhibit tissue-specific patterns depending on tumor origin, and non-neuroendocrine tumors might sometimes display focal expression. Moreover, CGA and SYP might be partially or totally absent in highly proliferative neuroendocrine carcinomas, making the diagnosis particularly challenging on small biopsies of metastatic lesions with unknown location of the primary tumor. The advent of second-generation neuroendocrine markers ISL LIM Homeobox 1 (ISL1), INSM Transcriptional Repressor 1 (INSM1) and Secretagogin (SECG) have expanded the pathology toolbox considerably, constituting markers that often retain expression even in poorly differentiated neuroendocrine carcinomas. As non-neuroendocrine tumors seldom express these antigens, the specificity of ISL1, INSM1 and SECG make them welcome additions to clinical practice. In this commentary, recent advances of this field as well as initial clinical experiences from a tertiary neuroendocrine center are discussed.

## 1. Introduction

### 1.1. First-Generation Neuroendocrine Markers

Although pathology laboratories are diverse in terms of the amount of immunohistchemical markers available for diagnostic purposes, the most commonly employed markers in the context of diagnosing a neuroendocrine tumor (NET) constitute Chromogranin A (CGA), Synaptophysin (SYP), Cluster of differentiation 56 (CD56) and Neuron-specific enolase (NSE). CGA is essential for the maturation secretory granules in many endocrine cells, and also regulate the exocytosis process [1]. Its use as an immunohistochemical marker for endocrine tumors was firstly established for pheochromocytoma, medullary thyroid carcinoma and parathyroid adenoma [2], and monoclonal antibodies for this purpose were subsequently produced [3]. On a similar note, following the identification of SYP as a presynaptic vesicle protein in various neuronal tissues [4], succeeding studies identified SYP expression in pheochromocytoma and paraganglioma as well as in pancreatic islet cells and NETs [5]. CD56 (a cell adhesion molecule) and NSE (a glycolytic enzyme) have also been found expressed in various NETs, but the specificity is suboptimal as expression has been reported in unrelated tumors [6,7,8,9,10]. Apart from these markers of neuroendocrine differentiation, immunohistochemistry detecting expression of various hormones related to specific NET types is helpful, such as serotonin in small intestinal and appendiceal NETs and calcitonin in medullary thyroid carcinoma [11]. 

### 1.2. Second-Generation Neuroendocrine Markers

ISL LIM Homeobox 1 (ISL1, also known as ISLET1) was originally found to be selectively expressed in neuroendocrine and neuronal cells, and has since been established as a transcription factor with important roles orchestrating pancreatic endocrine differentiation [12,13]. Specifically, ISL1 binds the insulin gene promoter and regulates insulin gene expression, thereby providing a crucial role for neuroendocrine cells of the Langerhans islets [12]. In subsequent studies, ISL1 has been shown to also be expressed in pancreatic, duodenal, rectal and colonic NETs, in addition to Merkel cell carcinoma, pheochromocytoma/paraganglioma and medullary thyroid carcinoma (Table 1 and Figure 1) [14,15,16,17]. Similarly, INSM Transcriptional Repressor 1 (INSM1) is a transcription factor implicated in neuroendocrine cell differentiation during embryonal stages [18,19]. From a pathology perspective, INSM1 has been established as a consistent NET marker, with positive immunoreactivity in pancreatic, pulmonary and gynecological NETs to name a few (Table 1, Figure 1) [17,20,21,22,23,24]. Finally, Secretagogin (SECG) is a calcium-binding protein originally thought to be selectively expressed in Langerhans islets [25,26]. However, succeeding analyses have revealed rather ubiquitous expression patterns among NETs and neuroendocrine carcinomas (NECs) (Table 1 and Figure 1) [17,27,28]. 

In all, these three proteins are functionally distinct from CGA and SYP, and have multiple roles unrelated to the vesicle transportation system. From a biological context, these markers may therefore be consistent in terms of expression even if the NET/NEC downregulates its secretory machinery as part of the dedifferentiation process. 

### 1.3. Experiences and Reflections from Clinical Implementation

In endocrine pathology, immunohistochemical markers are well-attributed for their diagnostic and prognostic features, not least in terms of neuroendocrine neoplasia [29,30,31,32]. However, most markers come with their limitations, and established proteins indicative of neuroendocrine differentiation are no exception to this rule. CGA, SYP, CD56 and NSE are widely employed in clinical pathology laboratories, and positive immunoreactivity for any of these markers is readily observed in most NETs. The reduced specificity however, makes the markers prone to falsely identify a non-NET as NET, especially in the hands of less experiences pathologists not used to these markers on a daily basis. Notably, quite a few tumor types may exhibit focal or diffuse SYP immunoreactivity, including adenocarcinomas of various origin, malignant melanoma, sarcomas and adrenal cortical tumors (Figure 2) [17,29,33,34,35,36]. The addition of ISL1, INSM1 and SECG has partly bridged this problem, as the amount of non-NET cases with positivity towards neuroendocrine markers of both the first and second generation are expected to be few [17]. As of this, the combination of CGA, SYP, ISL1, INSM1 and SECG could be particularly useful to rule out false positive cases (Figure 2). In a recent study in which the clinical application of ISL1 (clone EP283, Cell Marque, CA, USA), INSM1 (clone A-8, Santa Cruz Biotechnology, CA, USA) and SECG (clone 778518, R&D Systems, MN, USA) was assessed in a large cohort of NETs and non-NETs, the specificity of all three markers to detect neuroendocrine differentiation was comparable to those of CGA and SYP, while the sensitivity was somewhat lower (attributed to tissue-specific expressional patterns) [17]. Thus, the authors advocate that a combination of first- and second-generation neuroendocrine markers could constitute a highly sensitive and specific panel for clinical use.

Yet another clinical issue is the proper identification of poorly differentiated NECs with high proliferation counts, as these tumors not seldom show reduced expression of CGA and SYP (Figure 3 and Figure 4). Consulting previous studies, both ISL1 and INSM1 seem to be consistently expressed even in poorly differentiated pancreatic NECs, thereby providing valuable information in NECs with reduced or absent CGA or SYP immunoreactivity [14,17]. Additionally, for pulmonary NECs, INSM1 has been proposed as a more sensitive marker than conventional neuroendocrine markers of the first generation [14,24,37]. On a practical note, how would this translate to the clinical routine? A recent, real-life example from my institution could serve as an illustration; a core needle biopsy of a metastatic poorly differentiated tumor to the liver, in which radiology indicated a possibly primary tumor in the pancreatic tail. Using immunohistochemistry, we identified immunoreactivity towards keratins, Pancreatic and duodenal homeobox 1 (PDX1) and P53, and the Ki-67 labeling index was 90%. Moreover, focal CGA positivity was noted in 30-40% of tumor cells. Additional markers (SYP, CD56, NSE as well as markers of acinar cell differentiation) were negative. Although the focal CGA positivity may indicate a NEC, additional analyses using second-generation neuroendocrine markers pinpointed diffuse nuclear ISL1 and INSM1 immunoreactivity. Thus, the combination of CGA, SYP, ISL1 and INSM1 was helpful in order to safely establish the NEC diagnosis, and the patient was offered systemic treatment soon thereafter. 

An additional aspect of diagnosing metastatic NETs/NECs with unknown primaries is the fact that tissue-based expressional patterns not always are consistent. Pathologists may use CK7/CK20 as blunt tools followed by combinations of transcription factors PDX1, Caudal Type Homeobox 2 (CDX2), Thyroid Transcription Factor 1 (TTF1) and Paired Box 8 (PAX8) to identify the tissue of origin [11,38], but overlaps exist, and given the complex genetics of highly proliferative NECs, subsets of tumors may either loose or gain the ability to express one or several of these markers. It has been noted that ISL1, INSM1 and SECG also may exhibit tissue-specific patterns, thus potentially facilitating the proper identification of the primary site (Table 1) [17]. While most NETs/NECs are positive for ISL1, small intestinal and appendiceal NETs recurrently display negativity for this marker, while being positive for INSM1 and SECG [17,39] (Figure 3). Similarly, pheochromocytomas and abdominal paragangliomas may be ISL1 and INSM1 positive, but are most often negative for SECG [17] (Table 1, Figure 1). Moreover, ISL1, INSM1 and SECG have been shown to be consistently positive also in colorectal NETs, an entity which is recurrently negative for CGA [17,40].

The complexity of endocrine pathology is perhaps best exemplified by the NET/non-NET mixed tumor entity entitled “MiNEN”, in which a *bona fide* NET component as well as a non-NET population (both constituting at least 30% of the total tumor cell content) are visualized [11,41]. In this aspect, the “30% rule” (tumors with less than 30% of tumor cells positive for NET markers could be viewed as non-NETs with focal neuroendocrine differentiation) is potentially easier to interpret if second-generation neuroendocrine markers also are employed. 

## 2. Discussion

Despite the rapid development of next-generation sequencing techniques, immunohistochemistry still enjoys the top position as the most recurrently used adjunct histopathological methodology in clinical routine. Fast, cheap and easy to interpret when standardized, the method is standard practice in modern pathology laboratories. Visualizing immunohistochemical markers in cancer diagnostics is a clinical cornerstone with great diagnostic and prognostic utility, and expressional patterns might also guide oncologists to specific therapeutic alternatives. Moreover, information regarding the immunoreactivity of a specific protein could also triage patients for further genetic counseling. When diagnosing NETs, the pathologist is largely dependent on this technique to safely visualize neuroendocrine differentiation, as morphology alone is usually not sufficient to properly diagnose a tumor as “neuroendocrine”. Traditionally, first-generation markers such as CGA and SYP have been widely used and reproduced, and are still considered gold standard in endocrine pathology. However, non-NET mimics with focal neuroendocrine differentiation are not entirely uncommon, and highly proliferative NECs recurrently lose the ability to express CGA and SYP. Moreover, CGA and SYP offer little help to the diagnostician in terms of a tumor’s primary location if this is not known. To counter this, the inclusion of second-generation neuroendocrine markers ISL1, INSM1 and SECG could (1) increase the likelihood of correctly pinpoint NETs when a non-NET is among the potential differential diagnoses, (2) identify poorly differentiated pancreatic and pulmonary NECs exhibiting absent/focal CGA and SYP immunoreactivity and also (3) give important clues regarding the origin of metastatic cases when a primary tumor has not been established through radiology and/or clinical examinations. 

## 3. Conclusions

Pathologists in tertiary centers with high NET volumes are encouraged to establish second-generation neuroendocrine markers as part of the clinical routine arsenal in order to improve the diagnostic capability. 

## Figures and Tables

**Figure 1 biology-10-00874-f001:**
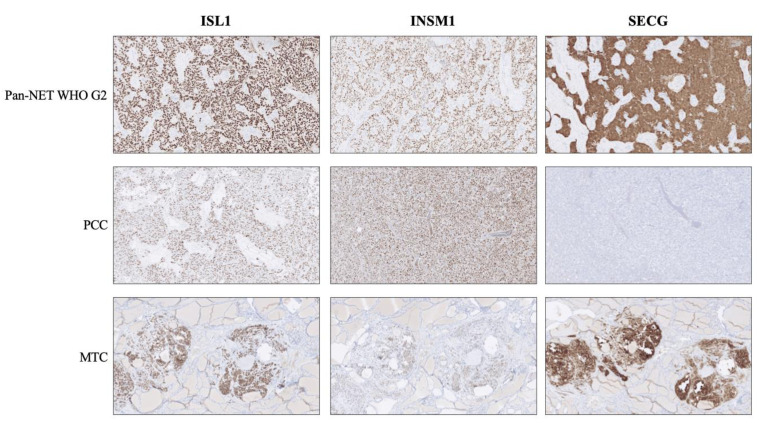
Expression of second-generation neuroendocrine markers ISL LIM Homeobox 1 (ISL1), INSM Transcriptional Repressor 1 (INSM1) and Secretagogin (SECG) in various neuroendocrine tumors. **Top row:** A pancreatic neuroendocrine tumor (Pan-NET) WHO grade 2 exhibiting distinct and diffuse nuclear immunoreactivity for all three markers, in addition to cytoplasmic SECG. **Middle row:** A pheochromocytoma (PCC) depicting the traditional staining pattern of this tumor type; positivity for ISL1 and INSM1, but absence of SECG. **Bottom row:** Medullary thyroid carcinoma (MTC) specimen with positive expression for all three markers. Note the absent staining in the surrounding thyroid parenchyma.

**Figure 2 biology-10-00874-f002:**
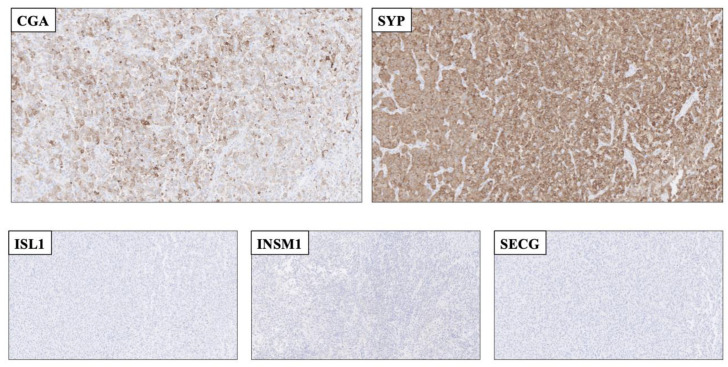
Application of second-generation neuroendocrine markers to rule out a neuroendocrine tumor as a plausible differential diagnosis. **Top row:** Resected cerebral metastasis of a tumor with unknown primary, with focal Chromogranin A (CGA) and diffuse Synaptophysin (SYP) staining. **Bottom row:** The metastasis was completely absent of ISL LIM Homeobox 1 (ISL1), INSM Transcriptional Repressor 1 (INSM1 and Secretagogin (SECG) immunoreactivity. As of this, the second-generation markers aided the pathologist in ruling out this lesion as a *bona fide* neuroendocrine tumor. The lesion was later diagnosed as a Ewing sarcoma exhibiting an *ESW1*-*ATF1* gene fusion (data not shown).

**Figure 3 biology-10-00874-f003:**
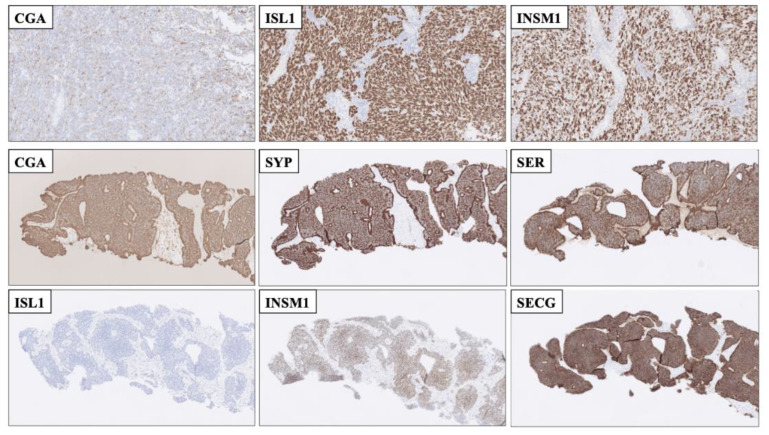
Immunohistochemical stainings displaying advantages of employing second-generation neuroendocrine markers compared to using first-generation markers alone. **Top row:** Resected cerebellar metastasis of a poorly differentiated neuroendocrine carcinoma (NEC) of unknown primary with only focal Chromogranin A (CGA) staining. The metastasis was diffusely positive for nuclear ISL LIM Homeobox 1 (ISL1) and INSM Transcriptional Repressor 1 (INSM1), Thus, the second-generation markers helped pinpoint this lesion as neuroendocrine. **Middle and bottom rows:** Core needle biopsy of a well-differentiated neuroendocrine tumor, metastatic to liver. First-generation CGA and Synaptophysin (SYP) were both positive, as was serotonin, but the second-generation marker ISL1 was negative, adjoined by INSM1 and SEGC positivity. This profile (ISL negative, INSM1 and SECG positive) is indicative of a small-intestinal neuroendocrine tumor, which was subsequently found on imaging.

**Figure 4 biology-10-00874-f004:**
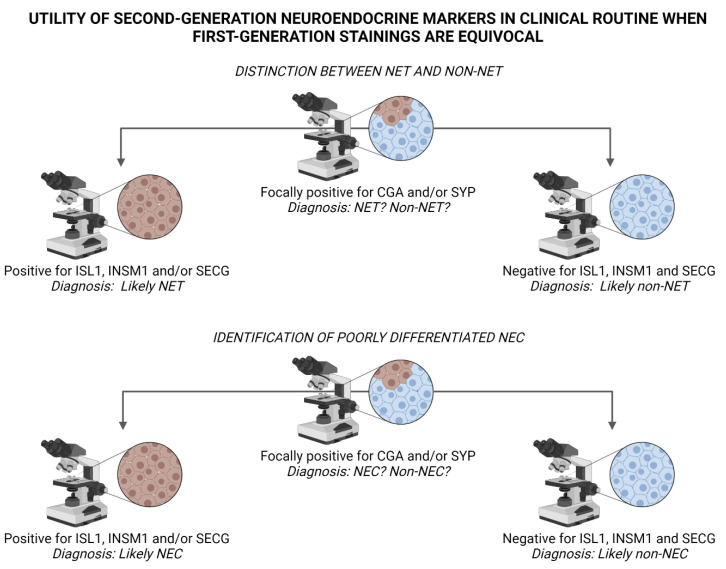
Suggested algorithm for interpreting staining outcomes of second-generation neuroendocrine markers ISL LIM Homeobox 1 (ISL1), INSM Transcriptional Repressor 1 (INSM1) and Secretagogin (SECG) when first-generation markers Chromogranin A (CGA) and Synaptophysin (SYP) are equivocal. **Top panel:** Diffuse positivity for ISL1, INSM1 and/or SECG indicate neuroendocrine differentiation, and could therefore argue in favor of a neuroendocrine tumor (NET) and help rule out the occurrence of a non-neuroendocrine tumor (non-NET) with focal neuroendocrine differentiation, for example adenocarcinoma, squamous cell carcinoma, malignant melanoma or sarcoma. **Bottom panel:** Similarly, a positive outcome when staining second-generation markers in poorly differentiated carcinomas may indicate a neuroendocrine carcinoma (NEC) as opposed to a non-neuroendocrine carcinoma (non-NEC). Image created using BioRender.com.

**Table 1 biology-10-00874-t001:** Schematic overview of tissue-specific expression patterns in NET for first- and second-generation neuroendocrine markers.

	CGA	SYP	ISL1	INSM1	SECG
Lung	+	+	+	+	+
Pancreas	+	+	+	+	+
Small intestine	+	+	-	+	+
PPGL	+	+	+	+	-
Colorectum	-	+	+	+	+

NET: neuroendocrine tumor; CGA: Chromogranin A; SYP: Synaptophysin; ISL1: ISL LIM Homeobox 1; INSM1: INSM Transcriptional Repressor 1; SECG: Secretagogin; PPGL: Pheochromocytoma/paraganglioma.

## Data Availability

Not applicable.

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
