# Peer review of "Second-Generation Neuroendocrine Immunohistochemical Markers: Reflections from Clinical Implementation"

_biology, 2021, doi:10.3390/biology10090874_

Round 1

Reviewer 1 Report

This commentary illustrates recent advances of using second-generation neuroendocrine immune-histochemical markers in the clinical setting. It presents useful insights of this filed as well as initial clinical experiences from a tertiary neuroendocrine center. 

Minor critiques:

  1. There is a typo in Figure 1 (labeled as Figure 2).
  2. In the original Figure 1, Ki-67 index was 90%, Chromogranin A (CGA) stain was negative and Synaptophysin (SYP) was reduced (data not shown). This data should be shown for the comparison purpose and the enhancement of clarity. 
  3. First-generation CGA and SYP were both positive (data not show), but the second-generation marker ISL1 was negative (C), adjoined by Secretagogin (SECG) positivity (D) (as well as immune-reactivity for INSM1, data not shown_. All the data mentioned in the legend should shown for the comparison purpose and the enhancement of clarity. 

Author Response

Chelsey Zhang

Editor, Biology (MDPI)   

Stockholm, 2nd of September 2021

Dear Dr. Zhang,

Please find attached my revised Commentary manuscript (biology-1354606) entitled “Second-generation neuroendocrine immunohistochemical markers: Reflections from clinical implementation”.

I would like to thank you for considering my submission and appreciate the time and efforts put down by you and the reviewers to provide me with comments that I think have improved the manuscript. All recommendations for revisions have been taken into consideration, and all changes in the manuscript are highlighted with Track Changes. Most notably, the revised version now contains four figures, of which two are novel, one is revised and one is left unaltered (but with new numbering). The specific replies to each reviewer suggestion are detailed below:

Reviewer #1:

This commentary illustrates recent advances of using second-generation neuroendocrine immune-histochemical markers in the clinical setting. It presents useful insights of this filed as well as initial clinical experiences from a tertiary neuroendocrine center.”

Reply: Thank you for these kind word.

Minor critiques:

“There is a typo in Figure 1 (labeled as Figure 2).”

Reply: Many thanks for noting this, this has been revised accordingly.

“In the original Figure 1, Ki-67 index was 90%, Chromogranin A (CGA) stain was negative and Synaptophysin (SYP) was reduced (data not shown). This data should be shown for the comparison purpose and the enhancement of clarity.”

Reply: I agree. The data was not shown as I did not have access to the performed CGA and SYP stains for this particular case. I have now added illustrative examples in the revised figure as suggested, using another tumor in which I had full access to several stains.

“First-generation CGA and SYP were both positive (data not show), but the second-generation marker ISL1 was negative (C), adjoined by Secretagogin (SECG) positivity (D) (as well as immune-reactivity for INSM1, data not shown_. All the data mentioned in the legend should shown for the comparison purpose and the enhancement of clarity.”

Reply: I concur. All stains were not available for scanning purposes, but the case was replaced – and subsets of the requested images have been added to the revised figure.

Reviewer #2:

“Very straigthforward commentary article. I would like to see the authors expanding with pictures all the immunoestainings and histopathological entities mentioned in their work. See my comments below:”

“Figure 1: Pictures A and B are the same picture. Please correct.”

Reply: Thank you for noting, this has been corrected in the revised figure.

“I would like to see more images of the entitities mentioned " In subsequent studies, ISL1 has been shown to also be expressed in pancreatic,duodenal, rectal and colonic NETs, in addition to Merkel cell carcinoma, pheochromocytoma/paraganglioma and medullary thyroid carcinoma . INSM1 has been found a consistent NET marker, with positive immunoreactivity in pancreatic, pulmonary and gynecological NETs.”

Reply: This could indeed be useful, and the resubmission now contains an additional figure (Figure 1) with some of these tumors featured with all three 2nd generation markers.

“It will be interesting to see  an example of  CGA, SYP, ISL1, INSM1 and SECG where was useful useful to rule out false posititvity.”

Reply: I agree. Images have been added to a novel Figure 2 in which false positivity was ruled out thanks to 2nd generation markers.

“Adding the most commonly antibodies clones used in routine clinical pathology would  be a nice addition.”

Reply: Another great suggestion. The clones used in my institution were added to the main text.

Thank you again for improving my manuscript substantially. I hope that the Editor and reviewers find the above-suggested changes in line with their intentions and will find this work of sufficient quality to warrant publication.

Best regards,

Carl Christofer Juhlin, MD, PhD

Karolinska Institutet, Stockholm, Sweden

Reviewer 2 Report

Very straigthforward commentary article. I would like to see the authors expanding with pictures all the immunoestainings and histopathological entities mentioned in their work.See my comments below:

  1. Figure 1: Pictures A and B are the same picture. Please correct.
  2. I would like to see more images of the entitities mentioned " In subsequent studies, ISL1 has been shown to also be expressed in pancreatic,duodenal, rectal and colonic NETs, in addition to Merkel cell carcinoma, pheochromocytoma/paraganglioma and medullary thyroid carcinoma . INSM1 has been found a consistent NET marker, with positive immunoreactivity in pancreatic,
    pulmonary and gynecological NETs.
  3. It will be interesting to see  an example of  CGA, SYP, ISL1, INSM1 and SECG where was useful useful to rule out false posititvity.
  4. Adding the most commonly antibodies clones used in routine clinical pathology would  be a nice addition .

Author Response

(The authors gave the same response as above.)
